

# Precipitation downscaling using a probability-matching approach and geostationary infrared data: An evaluation over six climate regions

Ruifang Guo[1,2], Yuanbo Liu[1], Han Zhou[1,2], Yaqiao Zhu[3]

[1]Key Laboratory of Watershed Geographic Sciences, Nanjing Institute of Geography and Limnology, Chinese Academy of Sciences, No. 73 East Beijing Road, Nanjing 210008, China
[2]University of Chinese Academy of Sciences, No. 19 Yuquan Road, Beijing 100049, China
[3]College of Urban and Environmental Sciences, Hubei Normal University, No.11 Cihu road,Huangshi 435002, China

*Correspondence to:* Yuanbo Liu (ybliu@niglas.ac.cn)

**Abstract.** Precipitation is one of the most important components of the global water cycle. Precipitation data at high spatial and temporal resolutions are crucial for basin-scale hydrological and meteorological studies. In this study, we proposed a cumulative distribution of frequency (CDF)-based downscaling method (DCDF) to obtain hourly 0.05 °×0.05 ° precipitation data. The main hypothesis is that a variable with the same resolution of target data should produce a CDF that is similar to the reference data. The method was demonstrated using the 3 hourly 0.25 °×0.25 ° Climate Prediction Center Morphing

method (CMORPH) dataset and the hourly 0.05 °×0.05 ° FY2-E Geostationary (GEO) Infrared  (IR) temperature brightness (Tb) data. Initially, power function relationships were established between precipitation rate and Tb for each 1 °×1 ° region. Then the CMORPH data were downscaled to 0.05 °×0.05 °. The downscaled results were validated over diverse rainfall regimes in China. Within each rainfall regime, the fitting functions coefficients were able to implicitly reflect the characteristics of precipitation. Qualitatively, the downscaled estimates were able to capture more details about rainfall

motions and changes. Quantitatively, the time series of the downscaled estimates were more similar to the rain gauge data than the original CMORPH product at the daily scale. The downscaled estimates not only improved spatio-temporal resolutions, but also performed better (Bias: -7.35%~10.35%; correlation coefficient (CC): 0.48~0.60) than the CMORPH product (Bias: 20.82%~94.19%; CC: 0.31~0.59) over convective precipitating regions. The downscaled results performed as well as the CMORPH product over regions dominated with frontal rain systems and performed relatively poorly over

mountainous or hilly areas where orographic rain systems dominate.

## 1 Introduction

Precipitation is a critical component in the global water cycle (Barrett and Martin, 1981; Smith et al., 1998; Tobler, 2004). Precipitation data at spatio-temporal resolutions are favoured mainly for two reasons. First, the poor representativeness and unevenly distribution of gauge stations make it incapable of reflecting spatially the precipitation variations (Hughes, 2006, 

Collischonn et al., 2008; Javanmard et al., 2010). Second, ground radar systems can provide full coverage spatial data for



most regions, but RADAR is very week in view of the precipitation intensity and subject to short time series. Moreover, the validation poses a big challenge for hydrological applications (Krajewski and Smith, 2002). These problems can be effectively resolved by using satellite remote sensing techniques.

A number of techniques have been developed to estimate or retrieve precipitation (Kidd and Levizzani, 2011). Based on these technologies, precipitation datasets have been produced at various resolutions, including the Global Precipitation Climatology Project (GPCP) (Huffman et al., 1997, 2001, 2009), the Tropical Rainfall Measuring Mission (TRMM) Multi-Satellite Precipitation (TMPA) (Huffman et al., 2007), the Climate Prediction Center Morphing method (CMORPH) (Joyce et al., 2004) and the Global Satellite Mapping of Precipitation (GSMaP) (Ushio et al., 2009), especially over the last 20 years. The typical spatial resolution of these products is 0.25 °×0.25 °(Dinku et al., 2007; Ebert et al., 2007; Hirpa et al., 2010; Sohn et al., 2010; Bitew and Gebremichael, 2011; Romilly and Gebremichael, 2011; Thiemig et al., 2012; Hu et al., 2014). This coarse resolution generally impedes the applications of the data for basin-scale hydrological and meteorological studies (Mekonnen et al., 2008). A downscaling procedure would therefore be highly necessary to meet the requirements of small-scale (<10 km) applications.

Downscaling approaches was first used to interpolate regional-scale atmospheric predictor variables to point-scale meteorological series (Karl et al., 1990; Wigley et al., 1990; Hay et al., 1991; 1992). Currently, downscaling approach is well developed and can be categorized into regression method, weather pattern approach, stochastic weather generator and limited-area climate modelling (Wilby and Wigley, 1997; Cannon, 2008). Most approaches are based on meteorological or climate models, and assume that relationships can be established between atmospheric parameters at disparate temporal and/or spatial scales (Giorgi and Mearns, 1999; Willems and Vrac, 2011; Kenabatho et al., 2012). At present, these methods are generally available to obtain precipitation data using ground-based data.

Various downscaling techniques have been developed to improve the resolution of satellite precipitation data. Immerzeel et al. (2009) used an exponential relationship between 1-km Normalized Difference Vegetation Index (NDVI) and precipitation to downscale TRMM 3B43 precipitation data on the Iberian Peninsula. Jia et al. (2011) used a linear regression relationship between a combination of NDVI/DEM and precipitation to downscale TRMM 3B43 precipitation data in the Qaidam Basin of China. Duan and Bastiaanssen (2013) used a two-degree polynomial regression model between NDVI and precipitation to downscale TRMM 3B43 precipitation data in the Lake Tana Basin, Ethiopia and the Caspian Sea Region, Iran. These studies manifest the potential of downscaling methods to obtain fine-resolution precipitation (<10km), while mainly focus on precipitation data with low temporal resolutions (i.e., annual or monthly).

The main objective of this study is to develop a regression-based downscaling method to obtain precipitation estimates with a high spatio-temporal resolution (0.05 °, hourly). Barrett et al. (1991) proposed a cumulative histogram method to relate precipitation observations to satellite estimates in an effort to avoid bias problems related to simple regression. In this study, we propose a cumulative distribution of frequency (CDF)-based downscaling method (DCDF) and perform a preliminary validation using CMORPH and geostationary (GEO) infrared (IR) temperature brightness (Tb) data. This new method can 1) lead to a better understanding of satellite precipitation data and 2) stimulate scientific interests to engender the development





of precipitation data with improved resolutions. The following section introduces the principle, framework and procedure of the downscaling method. Section 3 details the test areas and data processing. Section 4 presents the major findings followed by discussion in section 5. Finally, section 6 concludes.

## 2 Methodology

### 2.1 CDF matching

CDF matching is a probability based matching process. It assumes a variable (V) should produce a similar CDF to the reference variable (T). The frequencies of T and V are shown in Equations (1)-(2), and the cumulative frequencies in Eqs. (3)-(4).

$$P_t = f_1(t) \tag{1}$$

$$P_v = f_2(v) \tag{2}$$

$$C_t(t) = \int_{T_1}^{t} f_1(t)\,\mathrm{d}t \tag{3}$$

$$C_v(v) = \int_{V_1}^{v} f_2(v)\,\mathrm{d}v \tag{4}$$

where $P_t$ and $P_v$ are the frequency of T and V, $f_1(t)$ and $f_2(v)$ are probability density functions of T and V, and $C_t(t)$ and $C_v(v)$ are the cumulative density functions of T and V, respectively.

The steps for CDF matching are summarized in Fig.1. First, T and V are shown into histograms (Fig. 1a). The frequency of an arbitrary point $t_i$ (or $v_i$) on the $f_1(t)$ [or $f_2(v)$] curve can be expressed as $P(T=t_i)=f_1(t_i)$ [or $P(V=v_i)=f_2(v_i)$]. Second, these histograms are transformed into cumulative histograms (Fig. 1b). The cumulative frequency of an arbitrary point $t_i$ (or $v_i$) on the $C_t(t)$ [or $C_v(v)$] curve can be expressed as $C(T<t_i)=\int_{T_1}^{t_i} f_1(t)\,\mathrm{d}t$ [or $C(V<v_i)=\int_{V_1}^{v_i} f_2(v)\,\mathrm{d}v$].

Third, these cumulative histograms are matched so that V has a cumulative histogram similar to T. Lastly, the V-to-T relationship is established (Fig. 1c). Magnusson et al. (2015) demonstrated that CDF matching works better than histogram-matching method when low values have high frequencies, which is generally the case for precipitation.

### 2.2 Downscaling

Our method is based on the work of Barrett et al. (1991) and Kidd and Levizzani (2011). Rainfall can be inferred from IR imagery because heavy rainfall tends to be associated with large, tall clouds with cold cloud tops. Therefore, empirical relationships between precipitation rate and Tb are derived (Arkin and Meisner, 1987; Greene and Morrissey, 2000; Prigent, 2010). However, these relationships are indirect, and exhibit significant variations during the lifetime of a rainfall event. They also differ among rain systems and climatological regimes, which cause large uncertainties in precipitation estimations





(Kidd and Levizzani, 2011). Ba and Gruce (2001) demonstrated that a two-degree polynomial model is more effective for describing the relationship, and that the coefficients of the model are regional dependent. Overall, the precipitation-Tb relationship is highly variable over time and space.

Microwave (MW) radiation reflects the physical structures of clouds. Emission from rain droplets increase MW
radiation, and scattering by precipitating ice particles decreases MW radiation. Although MW techniques are physically more direct than those based on IR radiation, they both can reflect rainfall events. Therefore, we assume that IR signal produces a similar frequency distribution of precipitation rates to MW signal over a certain region during a certain period. Barrett et al. (1991) proposed a cumulative-histogram-matching method to relate rainfall observations to satellite precipitation data. Kidd et al. (2003) applied the same method to estimate rainfall using passive microwave (PMW) and IR
data over Africa.

The assumptions behind downscaling method include 1) Tb has similar cumulative frequency as precipitation rate at certain spatial and temporal scales, and 2) satellite precipitation products provide relatively accurate estimates with low spatial and temporal resolutions. In contrast, GEO-IR data have high spatio-temporal resolution yet with low accuracy. Illustrated in Fig. 2, the downscaling method explores the advantages of satellite precipitation product and GEO-IR data.
Specifically, 1) to aggregate Tb ($Tb_h$) from a high resolution to a low resolution ($Tb_l$) similar to the precipitation data (Eq. 5), and 2) to apply the CDF matching to the $Tb_l$ and precipitation rate ($R_l$) to obtain a $Tb_l$-$R_l$ relationship and a rain-no-rain threshold (Eq. 6). The downscaled precipitation rates are estimated based on the $Tb_l$-$R_l$ relationships (Eq. 7).

$$Tb_l = \frac{1}{n}\sum_{i=0}^{n} Tb_h(i) \qquad (5)$$

$$Tb_l = m \times R_l^{e} \qquad (6)$$

$$R_h = (\frac{Tb_h}{m})^{1/e} \qquad (7)$$

where $Tb_h$ denotes high-resolution GEO-IR Tb data. $Tb_l$ denotes upscaled Tb data. $R_l$ denotes the low-resolution precipitation product. $R_h$ denotes the derived high-resolution estimates. m and e are coefficients of the Tb-R relationship, and n is the number of high-resolution pixels within a low-resolution pixel.

Under the assumption that colder clouds are linked to higher rainfall than warmer clouds, the downscaling method
assumes a monotonically increasing precipitation rate with decreasing Tb. Therefore, cumulative histograms of precipitation rate and Tb are matched, so that the occurrence of the heaviest precipitation is associated with the Tb values linked to the heaviest rainfall. Decreasing Tb values are assigned to increasing precipitation rates so that the final distribution of Tb assigned to the precipitation rates is the same as that determined using precipitation rate data. In addition, the rain-no-rain threshold is the Tb value with the same cumulative frequency as that of the precipitation rate defined at the non-raining
frequency. That is, the rain-no-rain threshold is the maximum Tb value in the CDF matching procedure.

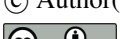



## 2.3 Validation

We adopted the validation strategies employed by Hirpa et al. (2010) and Chen et al. (2013). The Correlation coefficient (CC), root mean square error (RMSE) and bias are used to evaluate the results.

$$CC = \frac{\sum_{i=1}^{n}(G_i - G)(E_i - E)}{\sqrt{\sum_{i=1}^{n}(G_i - G)^2 \sum_{i=1}^{n}(E_i - E)^2}} \quad (8)$$

$$RMSE = \sqrt{\frac{1}{n}\sum_{i=1}^{n}(E_i - G_i)^2} \quad (9)$$

$$Bias = \frac{\sum_{i=1}^{n} E_i}{\sum_{i=1}^{n} G_i} - 1 \quad (10)$$

where $G_i$ denotes rain gauge precipitation, G represents the average of the gauge measurements, $E_i$ denotes an estimate with an average of E, and N is the total number of data pairs.

Categorical metrics are used to evaluate the ability to diagnose rainy or non-rainy events, including probability of detection (POD), false alarm rate (FAR) and Heidke skill score (HSS) (Table 1) (Wilks, 1995). POD measures the fraction of number of actual rainy events that are correctly reported as rainy events by the rainfall estimates, and ranges from 0 to 1. FAR measures the fraction of number of actual non-rainy events that are incorrectly reported as rainy events by the satellite estimates, and ranges from 0 to 1. HSS measures the fraction of the total number of cases accurately reproduced by the satellite estimates as rainy or non-rainy events relative to the fraction of correct random estimations, and ranges from -1 (totally negative skill) to 1 (perfect skill). HSS=0 denotes no skill relative to chance.

$$POD = \frac{a}{a+c} \quad (11)$$

$$FAR = \frac{b}{a+b} \quad (12)$$

$$HSS = \frac{2(ad - bc)}{(a+c)(c+d) + (a+b)(b+d)} \quad (13)$$

## 2.4 Variogram

Variogram describes how data correlates with distance. The variogram function γ(h) is defined as half of the mean value of the square of the difference between points separated by a distance h (matheron, 1963). The variogram can be described as

$$\gamma(x,h) = \frac{1}{2N(h)} * \sum_{i=1}^{N(h)} [E(x_i) - E(x_{i+h})]^2 \quad (14)$$



where N(h) is the number of data pairs with distance h, $E(x_i)$ and $E(x_{i+h})$ are the precipitation values at locations $x_i$ and $x_{i+h}$.

Variogram is generally an increasing function of distance h. The relationship between γ(h) and h is plotted as a variogram curve (Fig. 3). The relationship is commonly described using the nugget effect ($C_0$), sill ($C_0+C$) and range (D). $C_0$ denotes micro-scale variations, equated to of γ(0). $C_0+C$ denotes limit of the variogram γ (+∞). D denotes the distance at which the difference of the variogram from the sill becomes negligible. Variogram is used here to describe the spatial structure of satellite precipitation data.

# 3 Study areas and datasets

## 3.1 Study areas

Existing studies confirmed that the performances of satellite precipitation estimates are highly dependent on the rainfall regime (Arkin et al., 2006; Ebert et al., 2007; Gottschalck et al., 2005), which varies with climate zone, latitude, longitude and elevation. Thus, six $5\,°\times 5\,°$ regions in China were selected for validation (Fig. 4). Their corresponding geographic and climatic characteristics are listed in Table 2. These areas are distributed from south to north and from east to west, and they incorporate most rainfall regimes.

Among the six regions, regions SE, CE and NE are located in the eastern monsoon region. It is warm and rainy during the southeast monsoon in June-August, and cold and dry during the northwest monsoon in December-February. These three regions are featured by low-elevation hills and plains. Regions CW, NW and TP are located in the non-monsoon region with a continental climate. CW and NW belong to arid region, with 60%~70% precipitation occurring in June-August. CW has a relatively high elevation, mainly covered by plateaus, mountains and basins. NW is mainly covered by plateaus and basins. TP has a complex climate, mainly covered by plateaus and mountains. The seasonal precipitation distribution has two forms: a unimodal distribution in summer (June-August), and a bimodal distribution in spring (March-May) and autumn (September-November).

## 3.2 Datasets

### 3.2.1 Meteorological data

Rain gauge data were obtained from the National Meteorological Information Centre of the China Meteorological Administration (CMA) (http://cdc.cma.gov.cn/home.do). The datasets include daily precipitation records at 137 rain gauge stations in 2014 (Fig. 4). A strict quality control has been applied to check extreme values (Ma, 1998). There are 33, 29, 14, 31, 12 and 18 rain gauges in regions SE, CE, NE, CW, NW and TP, respectively. In the case of more than one station located within a pixel, the rain gauge values are averaged to represent the grid value. Statistical analyses were used to evaluate precipitation estimates at the daily scale. In addition, a disdrometer installed at Xingzi station (29.45 °N, 116.05 °E) in the



Jiangxi province (Fig. 4) provided hourly data in 2014, except June and July when the instrument was subject to a transmission error. Disdrometer data is used to evaluate the precipitation estimates.

### 3.2.2 Satellite data

IR data (10.7 μm) were collected from the Stretched Visible and Infrared Spin Scan Radiometer (S-VISSR) onboard FY2-E satellite. The data are available at National Satellite Meteorology Center (http://satellite.nsmc.org.cn/). FY2-E provides hourly coverage of eastern Asia from 75 °S to 75 °N. The IR Tb data were corrected for zenith angle viewing effects.

CMORPH is developed and produced by the Climate Prediction Center (CPC) in the National Oceanic and Atmospheric Administration (NOAA). CMORPH produces 0.25 °×0.25 ° 3 hourly global precipitation data using PMW and IR data. PMW data are from Microwave Imager (TMI) on TRMM, Special Sensor Microwave Imager (SSM/I) on Defense Meteorological Satellite Program (DMSP) satellites 13-15, Advanced Microwave Scanning Radiometer-Earth Observing System (AMSR-E) on Aqua, and Advanced Microwave Sounding Unit-B (AMSU-B) on NOAA satellite 15-18. Precipitation estimates are generated with the algorithms of Ferraro (1997) for SSM/I, Ferraro et al. (2000) for AMSU-B and Kummerow et al. (2001) for TMI. IR data are obtained from the GEO Operational Environmental Satellites (GOES) 8/10, European Meteorological Satellites (Meteosat) 5/7 and Japanese GEO Meteorological Satellites (GMS) 5. CMORPH uses motion vectors derived from GEO satellite IR imagery to propagate the relatively high quality precipitation estimates derived from PMW data (Joyce et al., 2004). Hence, quantitative precipitation estimates are based solely on PMW data. GEO-IR data are not used to estimate precipitation but rather to interpolate between two PMW-derived precipitation rate fields.

It seem that IR data are used twice, one for original CMORPH generation and the other for downscaling CMORPH. In fact, IR data serve as an intermediate variable for an interpolation purpose in the first step. While IR data serve as an ancillary variable in the second step for developing a precipitation-Tb relationship. The CMORPH product is essentially derived from MV observations, and therefore the use of IR data is reasonable.

We selected CMORPH as reference precipitation data instead of other High Resolution precipitation Products (HRPPs) for following reasons. This study focuses on developing a downscaling method and its application to low-resolution precipitation products to obtain fine-resolution precipitation. Thus, products with similar resolutions to GEO-IR data (0.05 °) are not used, such as CMORPH at 0.072 °, GSMaP at 0.1 ° and PERSIANN CCS at 0.04 °. TRMM 3B42 (RT) and Naval Research Laboratory Blended (NRLB) (Turk, 2005) algorithm combine MW-calibrated IR estimates. Thus, downscaling method would result in IR reusage.

### 3.3 Processing

The steps for data processing is shown in Fig. 5, including 1) FY2-E IR and CMORPH data collection, 2) database generation, 3) relationship building and 4) precipitation rate estimation. The first step was introduced in section 3.2, and hereafter, we describe the steps 2-4.

a. Generate the histogram database



IR-Tb data ($Tb_{0.05}$) were aggregated to a 0.25 ° grid ($Tb_{0.25}$) for each 3-hour period (0000-0300, 0300-0600, . . ., 2100-2400 UTC), in order to match the spatial and temporal resolutions as CMORPH. Then, IR-Tb ($Tb_{0.25}$) and CMORPH precipitation rate ($R_{0.25}$) were recorded in a database. The sample area for CDF matching was determined as follows. The horizontal and temporal scales of stratiform precipitation range from $10^1$ to $10^3$ kilometers and from hours to days (Orlanski, 1975; Trapp, 2013), while those of cumuliform precipitation range from a few km to tens of kilometers and from minutes to hours (Orlanski, 1975; Rickenbach, 2008). Based on previous studies (Kidd et al., 2003; Huffman et al. 2007), the downscaling procedure was conducted at 1 °×1 ° grids over a 10-days period. To reduce the heterogeneity among grids, a 3×3 window was used for smoothing purpose.

b. Build relationships between precipitation rate and Tb

The histograms of Tb and precipitation rate were generated and converted to cumulative histograms, and then matched using the CDF matching. Power function relationship between precipitation rate ($R_{0.25}$) and Tb ($Tb_{0.25}$) was established for each 1 °×1 ° area over a 10-days period. Meanwhile, various parameters, including a, b, rain-no-rain threshold and $R^2$, were obtained.

c. Estimate precipitation rate pixel by pixel at 1-hour, 0.05 °

Finally, the downscaled precipitation was obtained by applying power function and rain-no-rain threshold to Tb image ($Tb_{0.05}$).

## 4 Results

### 4.1 Tb-precipitation rate relationship

Fig. 6 shows fitting functions between the precipitation rate and Tb within each 1 °×1 ° grid. It was observed that Tb had a power function relationship with the precipitation rate. With an increase in the precipitation rate, Tb decreased, and the rate of change also reduced. The model fitting $R^2$ were all higher than 0.90. From the region SE to NE, the precipitation rate decreases, mainly subject to latitude. The maximum precipitation rate, rain-no-rain threshold and $R^2$ all showed decreasing trends. The maximum precipitation rate was 19.9 mm/h in region SE, 9.8 mm/h in region CE and 4.3 mm/h in region NE. The corresponding Tb values were 198 K, 202 K and 210 K, respectively, and the rain-no-rain threshold values were 265 K, 259 K and 249 K. The probability of precipitation rate was the largest for a given Tb in region SE, followed by region CE and then region NE. Regions CW and NW are arid, while TP is humid. The maximum precipitation rate was 3.5 mm/h for both region CW and NW and 11 mm/h for region TP. The rain-no-rain thresholds for regions CW and NW were approximately 230 K, while 254K for region TP. The probability of precipitation rate was the largest for a given Tb in region TP, because region TP has a complex rain system and high elevation. Generally, the fitting relationships reflected precipitation characteristics well.



## 4.2 Estimation results

Fig. 7 shows a comparison of the spatial distributions of CMORPH and DCDF precipitation estimates regions SE, NE and TP. The downscaled precipitation showed a similar spatial distribution to CMORPH, yet it reflected more detailed moving and changing processes of rainfall. To demonstrate clouds captured through DCDF and CMORPH, region SE was exemplified (14:00 to 16:00 June 21, 2014). Three cloud centers were observed in the southeastern and mid-eastern parts at 14:00. One hour later, two centers in the southeast moved eastward and joined together, while another center moved eastward. Two precipitation centers continued to move eastward at 16:00. In addition, D and $C+C_0$ values of DCDF (2.796 and 1.070) were higher than those of CMORPH (1.614 and 0.489). Large D and $C+C_0$ values indicate a high spatial dependence and high spatial variability. Thus, the spatial dependence and variability for high-resolution data were generally larger those for low-resolution data.

In region SE, clouds were relatively centralized with a high precipitation rate and were small in size. In region NE, clouds were discrete with a low precipitation rate and were widely distributed. In region TP, both centralized and discrete clouds appeared. Cumuliform cloud is the main type in region SE, while stratiform cloud is dominant in region NE, and both in region TP. Thus, the cloud distributions obtained through satellite data, especially using the DCDF approach, were consistent with the local characteristics. $C+C_0$ for cumuliform clouds was larger than that for stratiform clouds. A larger $C_0+C$ value was obtained for region SE (DCDF: 1.070; CMORPH: 0.489) than for region NE (DCDF: 0.007; CMORPH: 0.008). These results indicated that the DCDF method can reflect precipitation characteristics among rain systems and climatological regimes.

## 4.3 Validation

Fig. 8 shows a comparison among the DCDF, CMORPH and disdrometer at the hourly scale. The DCDF and CMORPH were able to capture rainfall events, although they differed in magnitude from the reference data in some cases. The DCDF effectively reflected the peak of each rainfall event, but could not exactly identify same starting and ending times of rainy events, resulting in somewhat delayed or advanced rainfall. The DCDF may detect non-rainy events as rainy events especially in dry seasons. CMORPH reported low-rain events as non-rainy events. Both of the DCDF and CMORPH estimates coincided with disdrometer data at precipitation rates ranging from 1 to 10 mm $\text{h}^{-1}$, such as the events from 10:00 to 14:00 on February 9 and from 21:00 on May 13th to 10:00 on May 14th.

To demonstrate performance of the DCDF method, a comparison of the DCDF and CMORPH estimates was conducted at the regional scale and at the point (rain gauge) scale. Fig. 9 shows the average precipitation of each region derived from rain gauge, DCDF and CMORPH. The daily average precipitation over each region showed almost identical temporal variations for DCDF and CMORPH. Both DCDF and CMORPH showed similar temporal patterns to the rain gauge observations, but they were probably subject to underestimation for regions CW and NW and overestimation for regions SE and TP. Fig.10 shows the time series data for each randomly selected rain gauge. The time series of DCDF were more



consistent with the rain gauge data than CMORPH at the point (gauge) scale, although the DCDF series were occasionally deviated from gauge data or misreported non-rainy events as rainy events. These results indicated that both DCDF and CMORPH demonstrated nearly equivalent performances at the regional scale, and the DCDF may perform better than CMORPH at the point (gauge) scale.

Table 3 lists the seasonal statistics for the six regions at the daily scale. Generally, DCDF performed better than CMORPH in region SE, while performed equivalently to CMORPH in regions CE and NE. Both of the DCDF and CMORPH showed better performances during rainy season. The DCDF generally showed the smallest biases between -7.35% and 10.35% (CC: 0.48~0.60) in region SE, and overestimated precipitation by 2.66%-33.95% (CC: 0.05~0.53) in regions CE and NE. CMORPH underestimated precipitation by 20.82%-94.19% (CC: 0.31~0.59) in region SE and showed biases between -93.2% and 6.78% (CC: 0.00~0.50) in regions CE and NE. Both the DCDF and CMORPH both exhibited bad performances in regions CW, NW and TP, and showed large biases (-73.75~2106%), low CC values (0.01~0.44) and high FAR values (0.33~1.00) during the winter. Further inspection showed that the DCDF overestimation was due to high POD and FAR, which may be caused by a low rain-no-rain threshold. The large biases for regions CW, NW and TP were likely due to insensitivity of precipitation data to very low precipitation in arid regions, and inability to estimate precipitation under complex rain systems.

## 5 Discussion

Existing downscaling methods involved an assumption that local scale patterns are driven by large-scale climatic fluctuations (Wilby and Wigley, 1997; Wilby et al., 2002). Most of these methods rely on meteorological or climate models, and utilize multiple parameters, such as temperature, humidity, pressure, vorticity and geostrophic airflow. These methods are not used to downscale satellite precipitation products possibly due to a diversity of parameters and complexity of the meteorological and climate models. In contrast, the DCDF method in this study assumes that IR retrieval should produce a frequency distribution of precipitation rates similar to that produced by of MW retrievals over a certain region during a certain period. That is, IR estimations and MW retrievals from clouds have strong statistical frequency similarities.

Due to high spatial and temporal variability of precipitation, the DCDF method must be conducted over a certain region during a certain period. The area and time period must be large enough for a reasonable sample size, but small enough to represent local characteristics. In the TMPA algorithm, relationship between IR and precipitation rate is built within a $1°×1°$ area by $3×3$ windows over the period of a month (Huffman et al. 2007). Kidd et al. (2003) obtained the relationship within $1°×1°$ area with the use of a $5°×5°$ Gaussian filter over a period of 5 days. Based on the horizontal and temporal scales of stratiform and cumuliform precipitation (Orlanski, 1975; Rickenbach, 2008, Trapp, 2013) and previous studies (Kidd et al., 2003; Huffman et al. 2007), the DCDF method is applied within a $1°×1°$ area by $3×3$ windows over a 10-day period. Nevertheless, the same gridded sample area is not the optimal selection. The size of sample area is determined according to local cloud type, and varies over space and time. It likely is our future work to improve precipitation estimates algorithm.




The DCDF method has two main disadvantages. The physical premise of the DCDF method is that cloud top temperature in the IR imagery is a simple empirical function of cloud top height, and that heavier rainfall tends to be associated with larger, taller clouds with colder cloud tops. Unfortunately, not all cold clouds precipitate, and precipitation does not always fall from cold clouds only (Barrett, 1970). This phenomenon results in misreporting. In addition, the rain-no-rain threshold is very critical for final precipitation estimates. The size of the sample area and the indirect relationship between IR-Tb and precipitation rate both affect the rain-no-rain threshold. However, both of them have uncertainties among rain systems and climatological regimes, resulting in uncertainties of rain-no-rain threshold.

Rain-gauge measure represents a space for a very small area while satellite precipitation products have a spatial resolution of several kilometers or more. Thus, high-resolution data is generally more similar to gauge data than low-resolution data. Furthermore, the characteristic scale is small for convective systems, and large for frontal rain systems. Convective precipitation dominates in region SE, while frontal rain system dominates in regions CE and NE. Thus, a rain gauge measure can represent a space for a smaller area in region SE than in regions CE and NE. Therefore, discrepancies between rain gauge observations and satellite estimates are lower in region SE than in regions CE and NE. CMORPH performed poorly in regions NW and TP, where orographic rain systems dominate (Hirpa et al., 2010; Romilly and Gebremichael, 2011; Gao and Liu, 2013). Our results are consistent with these findings.

## 6 Conclusions

Precipitation data with high spatial and temporal resolutions are highly needed in basin-scale hydrological and meteorological studies. Based on the works by Barrett et al. (1991) and Kidd and Levizzani (2011), this study proposed a DCDF method to obtain precipitation data at the hourly, 0.05° scale. The method was demonstrated using the CMORPH dataset and FY2-E GEO-IR Tb data in 2014. With the establishment of a power function relationship, improved precipitation estimates at the hourly and 0.05° resolution were produced. The DCDF precipitation estimates were validated using rain gauge data at six 5°×5° regions with different climate and geographical conditions in China.

There are three key points of the DCDF method. First, it explores the advantages of satellite precipitation estimates and GEO-IR data. The DCDF method assumes a monotonically decreasing Tb rate with an increase of precipitation rate, and that Tb data have the same cumulative frequency as that of precipitation rate for certain spatial and temporal scales. Second, the sample area where the CDF matching was conducted needs to be large enough for a reasonable sample size, but small enough to represent the local characteristics. In this study, size of the sample area was 1°×1° grid over a 10-day period based on the characteristic scale of precipitation clouds. (3) A power function relationship between precipitation rate and Tb was established for each sample area. Meanwhile, a rain-no-rain threshold was obtained as the Tb value with the same cumulative frequency as that of precipitation rate defined as non-rainy frequency. Generally, the threshold was the maximum Tb in the CDF matching procedure.



The established fitting relationships generally reflected the precipitation characteristics well in the six validation regions. For the distributions of precipitation clouds, the DCDF precipitation estimates showed a similar spatial distribution to that produced by CMORPH, but it reflected more detailed moving and changing processes of rainfall. The DCDF method can effectively reflect the precipitation characteristics among rain systems and climatological regimes. At the hourly scale, both

DCDF and CMORPH coincided with the disdrometer data at precipitation rates ranging from 1 to 10 mm h$^{-1}$. The DCDF effectively reflected the peak of each rainfall event, but could not exactly identify the starting and ending times of rainy events. The DCDF may detect non-rainy events as rainy events especially in dry seasons, while CMORPH reported low-rain events as non-rainy events. At the daily scale, DCDF and CMORPH had nearly equivalent performances at the regional scale, and DCDF may perform better than CMORPH at the point (rain gauge) scale. Generally, the DCDF performed better (bias:

7.35%~10.35%; CC: 0.48~0.60) than the original CMORPH product (bias: 20.82%-94.19%; CC: 0.31~0.59) over the regions where convective precipitation dominates. It performed as well as the CMORPH product over the regions where frontal rain systems dominate, and relatively poorly over mountainous or hilly areas, where orographic rain systems dominate.

*Competing interests.* The authors declare that they have no conflict of interest.

*Acknowledgments and data*

This work was partially supported by the State Key Program of the National Natural Science of China under Grant 41430855 and by the National High Technology Research and Development Program under Grant 2013AA12A301. The authors would like to thank Prof. Kidd (Dr Chris Kidd) for providing a report of SSM/I rainfall algorithms, and Prof. Xie (Dr Pingping Xie) for his guidance at the University of Maryland. The data used to produce the results of this paper may be obtained by

20 contacting the corresponding author.

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



**Table 1 Contingency table for the definition of the categorical metrics (Gebremichael, 2010)**

| Estimated precipitation | Reference precipitation | | Outcome |
|---|---|---|---|
| | Yes | No | |
| Yes | a | b | a+b |
| No | c | d | c+d |
| Outcome | a+c | b+d | n=a+b+c+d |

**Table 2 Geographic and climatic situations of the six regions in China**

| Region | Longitude latitude | Elevation rang(m) | Annual Precipitation (mm) | Climate zone | |
|---|---|---|---|---|---|
| SE | 110 °E~115 °E 23 °N~28 °N | 22~1405 | 1230 | Subtropical humid | Monsoon |
| CE | 114 °E~119 °E 33 °N~38 °N | 6~1533 | 670 | Warm temperate semi-humid | Monsoon |
| NE | 121 °E~126 °E 46 °N~51 °N | 147~740 | 460 | Mid temperate humid | Monsoon |
| CW | 99 °E~104 °E 34 °N~39 °N | 1368~8500 | 40 | Warm temperate arid | Non-monsoon |
| NW | 82 °E~87 °E 41 °N~46 °N | 320~2458 | 70~140 | Mid-temperate arid | Non-monsoon |
| TP | 89 °E~94 °E 28 °N~33 °N | 3552~8260 | 420 | Temperature plateau | Non-monsoon |

**Table 3 Validation results of the daily precipitation for CMORPH and DCDF in 2014 in the six study regions.**

| Indexes | Time | Type | SE | CE | NE | CW | NW | TP |
|---|---|---|---|---|---|---|---|---|
| B(%) | 1year | CMORPH | -29.60 | -12.82 | -7.09 | -5.57 | 120.22 | 26.41 |
| | | CDF | -3.91 | 11.54 | 15.85 | 32.82. | 145.43 | 52.33 |
| | SP | CMORPH | -20.82 | -3.31 | -45.50 | 45.44 | 159.02 | 83.32 |
| | | CDF | -7.35 | 2.94 | 31.23 | 50.92 | 191.79 | 100.36 |
| | SU | CMORPH | -22.12 | 3.17 | 6.78 | -43.92 | 143.11 | -9.49 |
| | | CDF | -10.47 | 2.66 | 5.94 | 25.91 | 217.04 | 7.53 |
| | FA | CMORPH | -57.75 | -33.00 | -16.90 | 10.90 | 114.44 | 43.22 |
| | | CDF | 5.92 | 33.95 | 19.88 | 25.78 | 128.51 | 59.77 |
| | WI | CMORPH | -94.19 | -32.83 | -96.20 | 1042 | -73.75 | 1655 |
| | | CDF | 10.35 | 20.54 | 22.39 | 1874 | 54.58 | 2106 |
| RMSE | 1year | CMORPH | 12.20 | 6.69 | 6.71 | 3.85 | 2.32 | 4.50 |
| | | CDF | 7.94 | 4.38 | 5.16 | 4.74 | 3.96 | 6.08 |
| | SP | CMORPH | 16.23 | 4.79 | 3.13 | 2.81 | 2.41 | 2.70 |
| | | CDF | 11.81 | 7.32 | 2.77 | 2.80 | 3.09 | 3.45 |
| | SU | CMORPH | 16.61 | 10.25 | 12.39 | 5.74 | 3.38 | 7.43 |
| | | CDF | 13.83 | 10.95 | 10.64 | 6.98 | 5.13 | 10.27 |
| | FA | CMORPH | 6.14 | 6.90 | 3.89 | 3.93 | 1.94 | 3.46 |
| | | CDF | 0.19 | 6.44 | 2.67 | 4.51 | 3.72 | 3.98 |
| | WI | CMORPH | 3.80 | 1.59 | 0.68 | 1.61 | 0.65 | 2.45 |
| | | CDF | 2.86 | 2.05 | 0.41 | 2.47 | 1.14 | 3.49 |
| CC | 1year | CMORPH | 0.52 | 0.32 | 0.32 | 0.17 | 0.33 | 0.28 |
| | | CDF | 0.60 | 0.47 | 0.42 | 0.29 | 0.29 | 0.33 |
| | SP | CMORPH | 0.59 | 0.34 | 0.36 | 0.17 | 0.07 | 0.04 |
| | | CDF | 0.66 | 0.40 | 0.38 | 0.17 | 0.05 | 0.04 |
| | SU | CMORPH | 0.36 | 0.19 | 0.25 | 0.17 | 0.40 | 0.23 |
| | | CDF | 0.48 | 0.26 | 0.46 | 0.44 | 0.44 | 0.37 |
| | FA | CMORPH | 0.40 | 0.50 | 0.36 | 0.07 | 0.32 | 0.11 |
| | | CDF | 0.52 | 0.53 | 0.46 | 0.10 | 0.21 | 0.08 |
| | WI | CMORPH | 0.31 | 0.02 | 0.00 | 0.05 | 0.03 | 0.06 |
| | | CDF | 0.52 | 0.17 | 0.05 | 0.01 | 0.02 | 0.15 |





| | | | | | | | | |
|-----|-------|--------|-------|-------|-------|-------|-------|-------|
| POD | 1year | CMORPH | 0.64 | 0.59 | 0.51 | 0.76 | 0.52 | 0.80 |
| | | CDF | 0.77 | 0.74 | 0.62 | 0.80 | 0.69 | 0.87 |
| | SP | CMORPH | 0.68 | 0.52 | 0.45 | 0.82 | 0.51 | 0.70 |
| | | CDF | 0.80 | 0.66 | 0.60 | 0.95 | 0.63 | 0.72 |
| | SU | CMORPH | 0.86 | 0.69 | 0.78 | 0.82 | 0.80 | 0.91 |
| | | CDF | 0.99 | 0.85 | 0.91 | 0.87 | 0.90 | 1.00 |
| | FA | CMORPH | 0.50 | 0.67 | 0.46 | 0.71 | 0.80 | 0.72 |
| | | CDF | 0.65 | 0.75 | 0.59 | 0.84 | 0.92 | 0.89 |
| | WI | CMORPH | 0.22 | 0.19 | 0.00 | 0.28 | 0.59 | 0.14 |
| | | CDF | 1.00 | 1.00 | 1.00 | 1.00 | 1.00 | 1.00 |
| FAR | 1year | CMORPH | 0.30 | 0.63 | 0.48 | 0.65 | 0.76 | 0.65 |
| | | CDF | 0.35 | 0.59 | 0.55 | 0.72 | 0.81 | 0.64 |
| | SP | CMORPH | 0.17 | 0.76 | 0.63 | 0.71 | 0.85 | 0.78 |
| | | CDF | 0.21 | 0.70 | 0.73 | 0.81 | 0.92 | 0.85 |
| | SU | CMORPH | 0.31 | 0.53 | 0.36 | 0.33 | 0.68 | 0.30 |
| | | CDF | 0.43 | 0.52 | 0.41 | 0.57 | 0.79 | 0.38 |
| | FA | CMORPH | 0.46 | 0.52 | 0.58 | 0.69 | 0.68 | 0.73 |
| | | CDF | 0.48 | 0.58 | 0.66 | 0.89 | 0.66 | 0.91 |
| | WI | CMORPH | 0.54 | 0.90 | 1.00 | 0.96 | 0.76 | 0.99 |
| | | CDF | 0.61 | 0.95 | 1.00 | 1.00 | 0.97 | 1.00 |
| HSS | 1year | CMORPH | 0.43 | 0.25 | 0.34 | 0.08 | 0.14 | 0.13 |
| | | CDF | 0.39 | 0.31 | 0.35 | 0.14 | 0.08 | 0.09 |
| | SP | CMORPH | 0.41 | 0.13 | 0.23 | 0.00 | 0.09 | 0.01 |
| | | CDF | 0.44 | 0.21 | 0.29 | 0.03 | 0.11 | 0.07 |
| | SU | CMORPH | 0.38 | 0.29 | 0.40 | 0.25 | 0.17 | 0.22 |
| | | CDF | 0.32 | 0.35 | 0.37 | 0.33 | -0.08 | 0.16 |
| | FA | CMORPH | 0.38 | 0.39 | 0.27 | -0.02 | 0.17 | 0.04 |
| | | CDF | 0.39 | 0.48 | 0.34 | 0.01 | -0.06 | 0.01 |
| | WI | CMORPH | 0.14 | 0.00 | -0.01 | -0.06 | 0.16 | -0.06 |
| | | CDF | 0.21 | 0.07 | 0.03 | -0.11 | 0.29 | -0.16 |





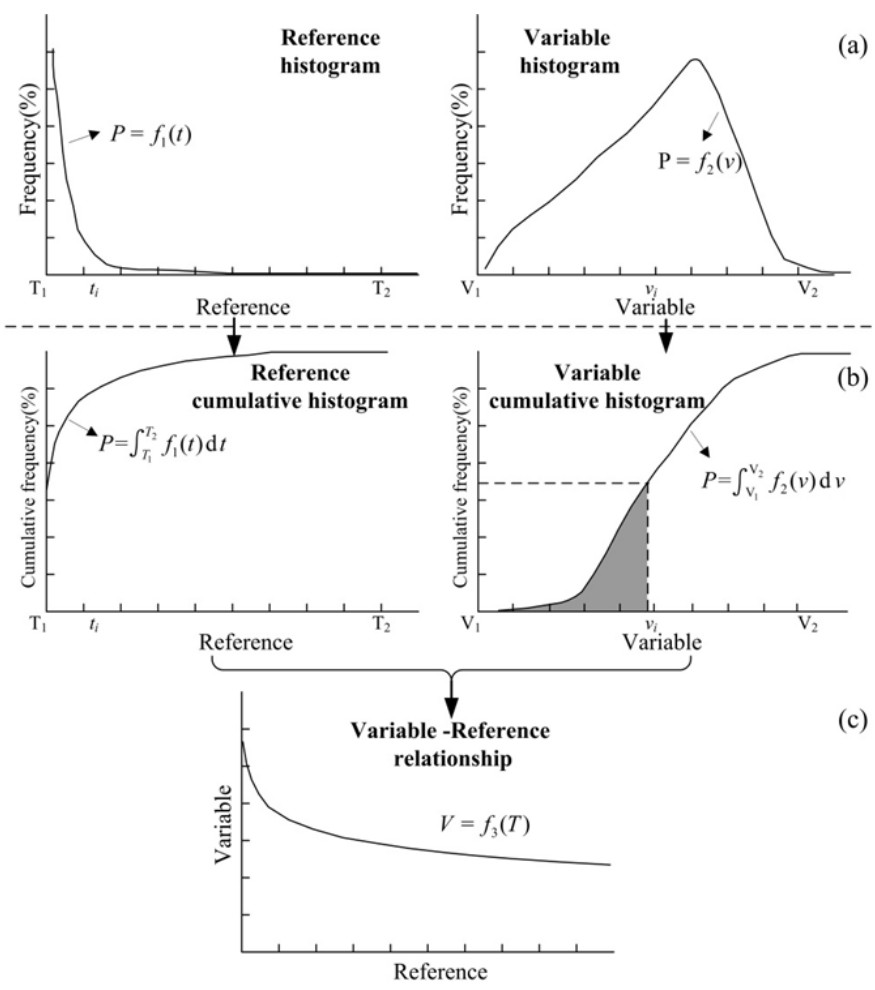

**Fig. 1. Schematic of the cumulative distribution of frequency (CDF) matching method**

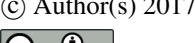



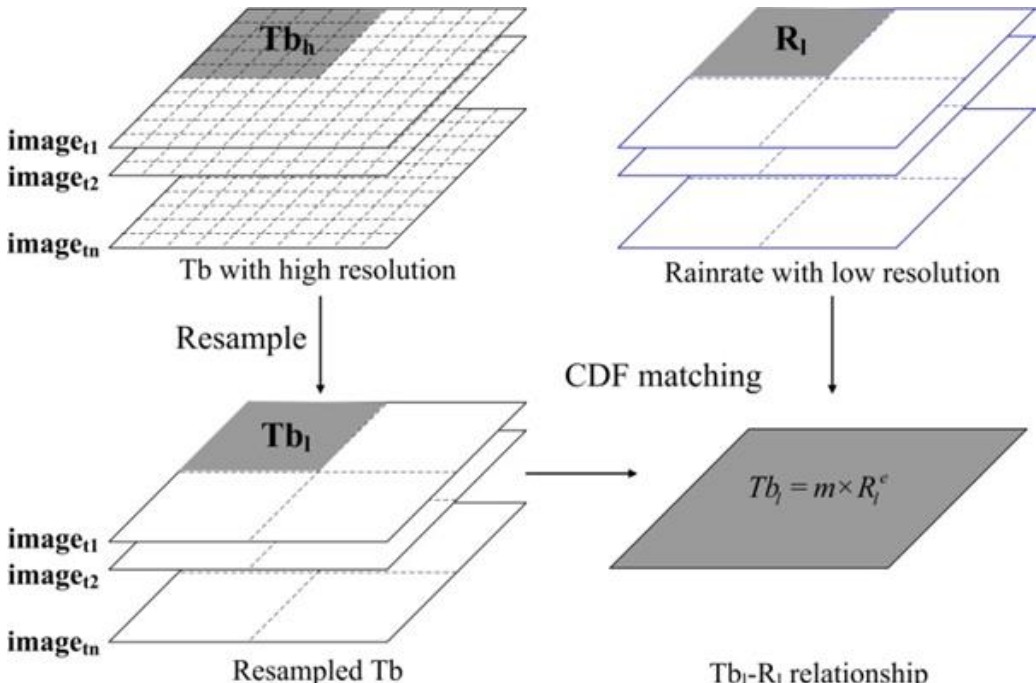

**Fig. 2. Schematic of the CDF-based downscaling method (DCDF) in this study**

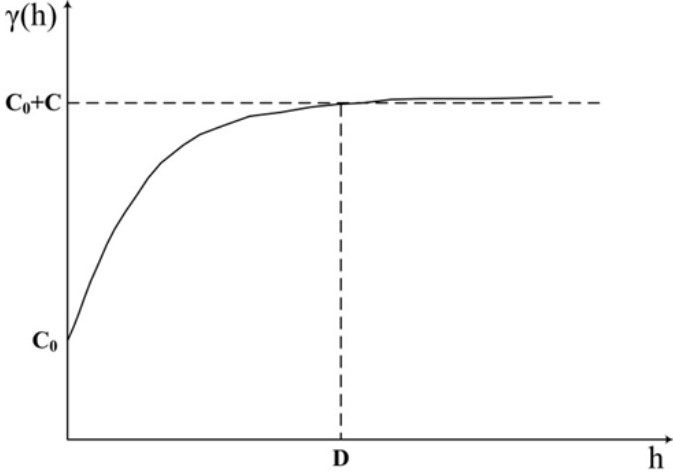

**Fig. 3. Schematic of the variogram curve showing nugget effect ($C_0$), sill ($C_0+C$) and range (D).**





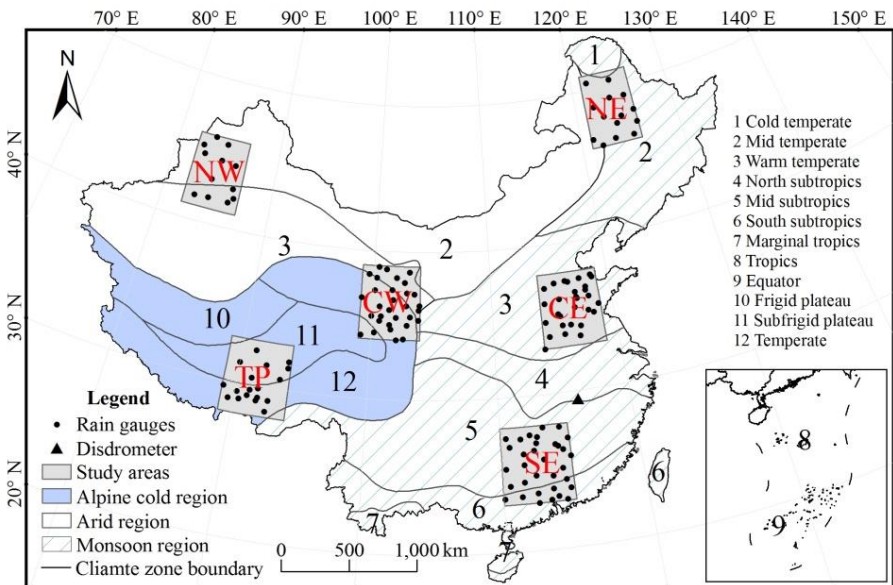

**Fig. 4. Geographic and climate situations of the six regions in China. The locations of the rain gauges are superimposed**

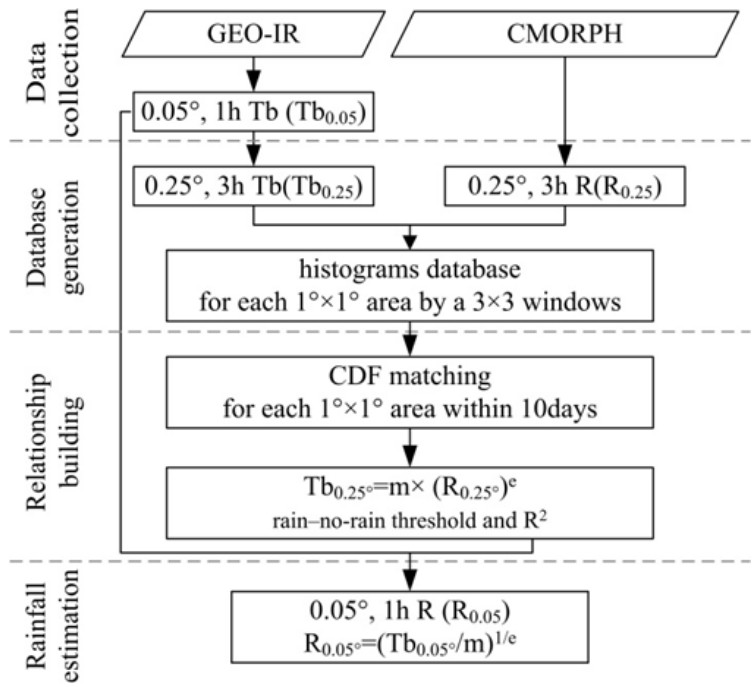

**Fig. 5. Flowchart of data processing for the DCDF method developed**





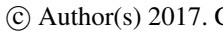

**Fig. 6. Examples of fitting of precipitation rate and Tb for each region in China during 20140709-20140718 for subregion SE (115°35′E, 27°28′N), subregion CE (115°39′E, 36°14′N), subregion NE (124°20′E, 51°42′N), subregion CW (101°38′E, 37°31′N), subregion NW (85°43′E, 46°47′N), and subregion TP (91°06′E, 30°29′N).**





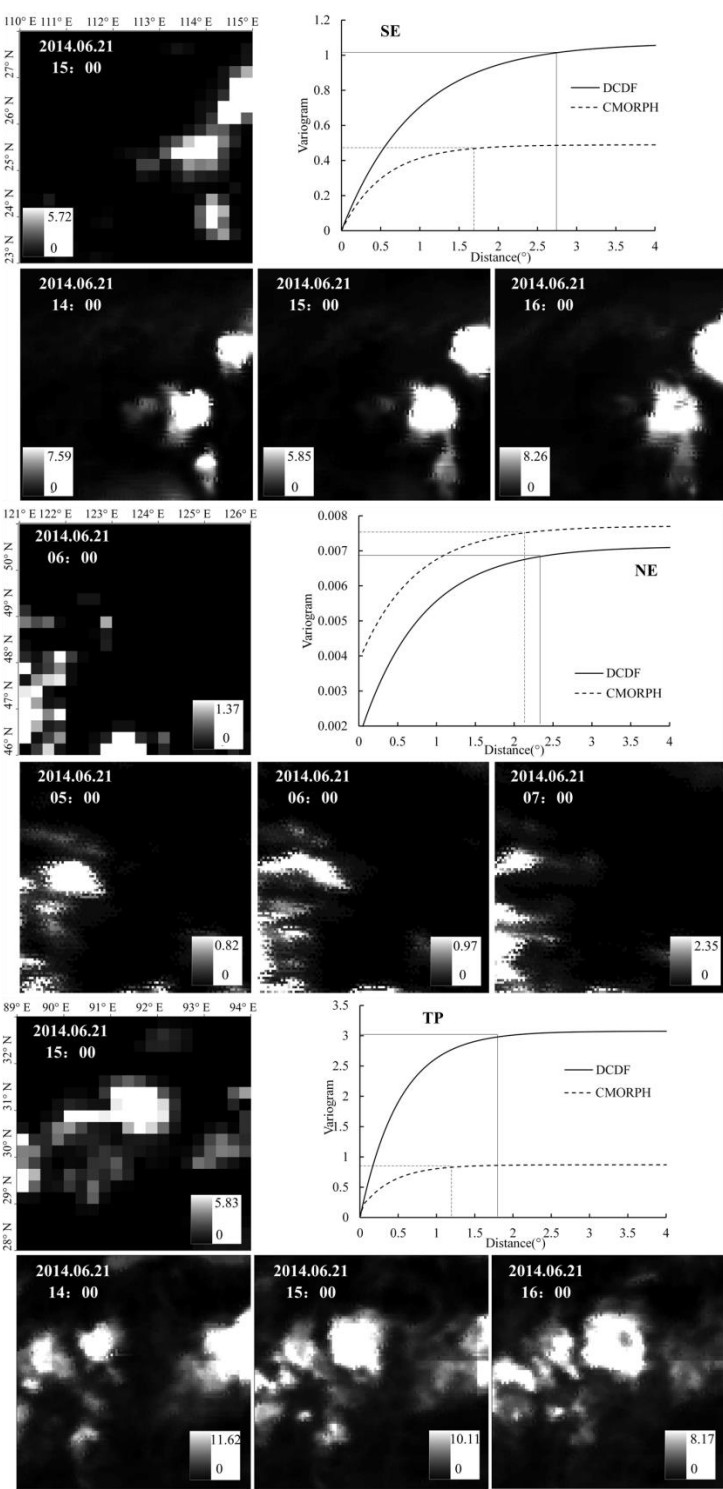

**Fig. 7.** CMORPH precipitation estimates at a nominal resolution of 0.25 ° and DCDF precipitation maps at a 0.05 ° resolution for regions SE, NE and TP







**Fig. 8. Time series of disdrometer data, original CMORPH and DCDF precipitation at hourly scale in 2014**





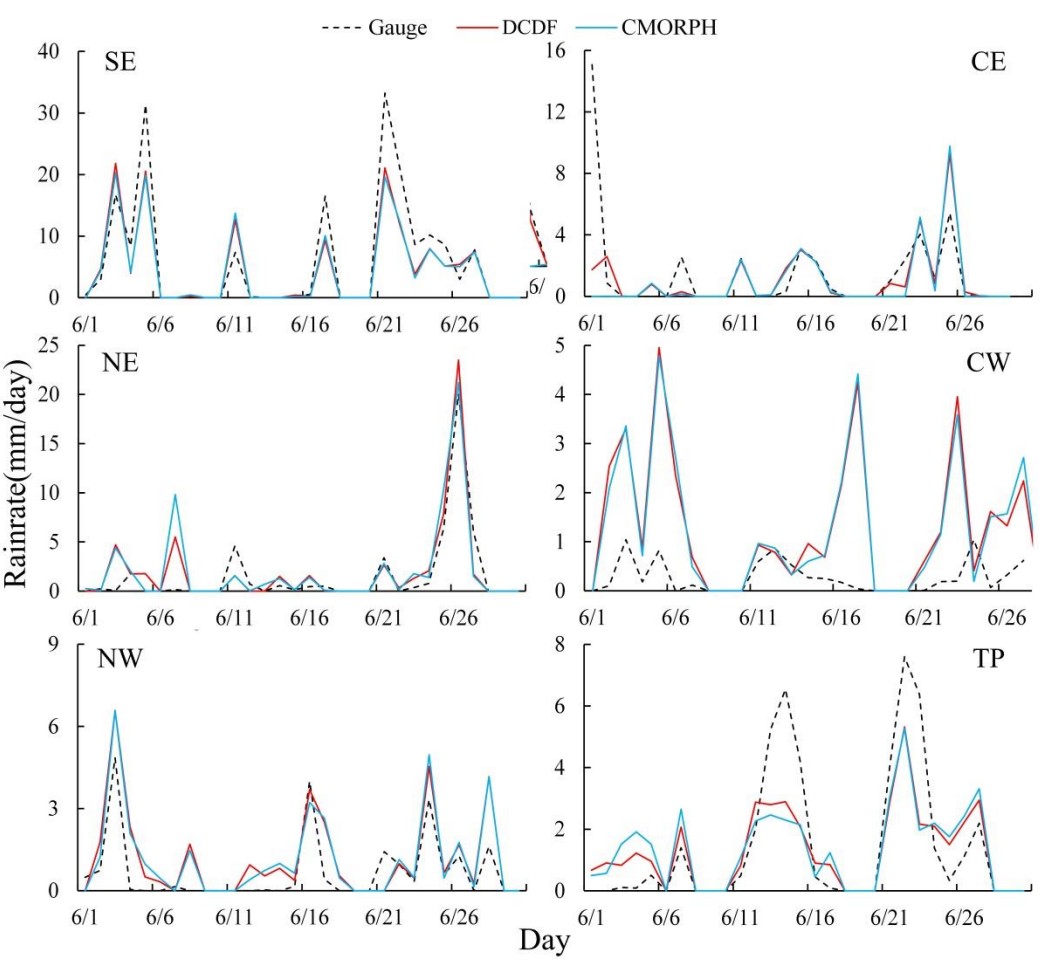

Fig. 9. Time series of the average precipitation of each region derived from gauge, DCDF and CMORPH at the daily scale in June 2014.





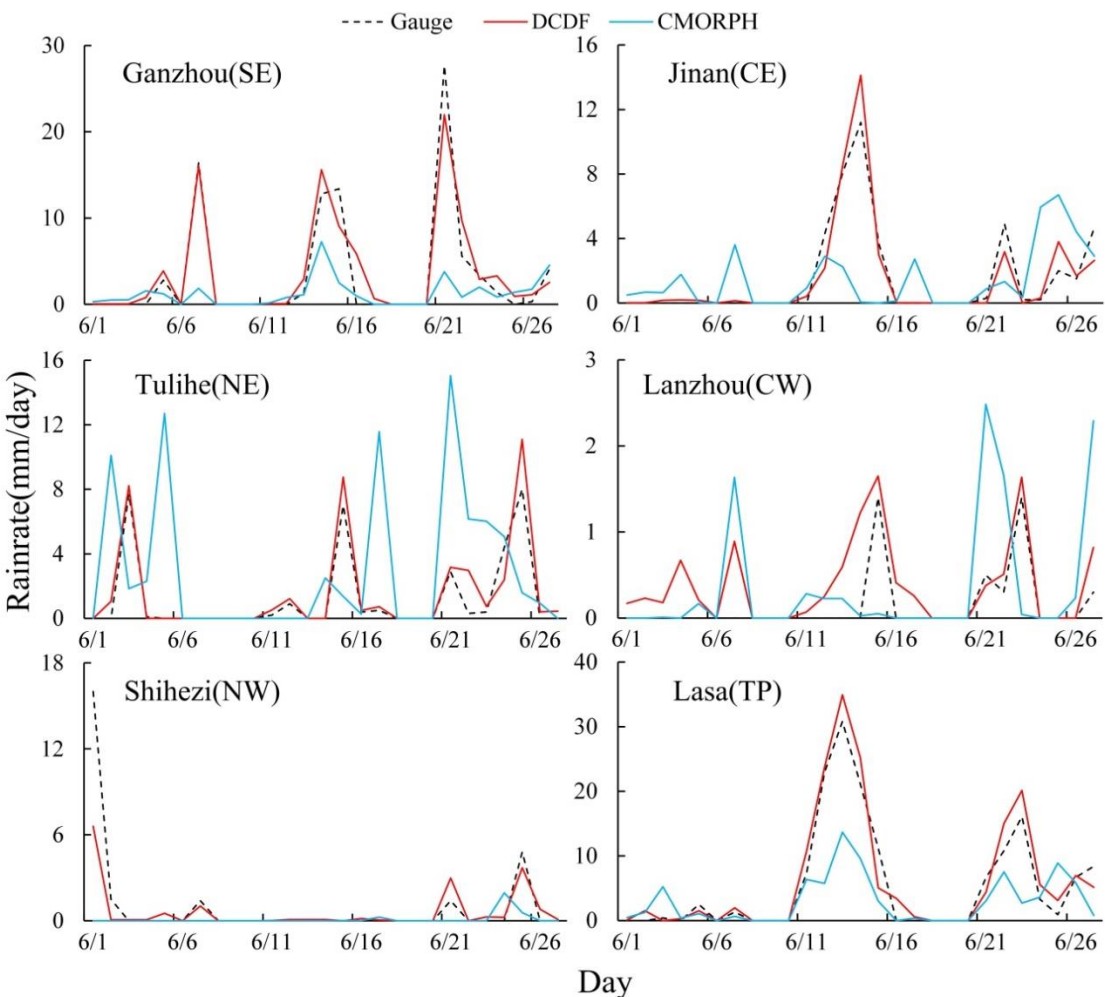

**Fig. 10. Time series of rain gauge data, original CMORPH and DCDF precipitation for each randomly selected gauge. Ganzhou station (SE), 113.1667 °E, 25.8667 °N; Jinan station (CE), 117.05 °E, 36.6 °N; Tulihe station (NE), 121.6833 °E, 50.4833 °N; Lanzhou station (CW), 103.8833 °E, 36.05 °N; (e) Shihezi station (NW), 86.05 °E, 44.3167 °N; (f) Lasa station (TP), 91.1333 °E, 29.6667 °N.**