# Peer review of "Precipitation downscaling using a probability-matching approach and geostationary infrared data: An evaluation over six climate regions"

_Hydrology and Earth System Sciences, 2017_

## Referee Comment (RC1) · Anonymous Referee #1 · 13 Dec 2017

here are my comments:

p1l11: "proposed" better present tense?

p2l2: which challenges and problems?

fig1: c is unclear what can be seen there

p3l19ff: the matching is not decribed, what means matching how is it done? Formula? Is that Quantile mapping?

p4l11: ..behind [the] downscaling..

p4l28ff: sentence is unclear, what is a non-raining frequency?

p5l20 and p6l2: A variogram

p5l21: (Matheron,..)

comment: Chapter 2.3 and 2.4 also fig 3, tab.1 are well know scores and techniques maybe you skip them.

Fig4: please exclude the islands with climate situations 8 and 9 from the map. the status of these territories are unclear.

Fig5: is again a processing scheme, maybe it is better to make one out of fig2 and fig5, if you change the order of chapter 2 and 3 you can combine chapters 3.3 and 2.2

p7: I found to much information in chapter 3.2.2 maybe you short it and only refer to the 2 or 3 most important references.

p8l15: [the] image ....definitely needs language editing

fig7: inscribe, which picture is cmorph data and which dcdf, and which picture belongs to which region, maybe confusing or unclear for the reader.

p9l7ff: please write sill, range when necessary instead of d and c+c0

p9-10l32ff: Fig 10, according to the shown events the conclusion is not significant, the better fit of dcdf at gauge scale may be pure luck.

p10l11ff: the bad performance of the approach in winter was something i except reading your methods. How are the correlations between tb and rain in teh winter months?

p10l13: rain-no-rain threshold , where is this threshold defined? how large is threshold?

p10l15: what is meant by complex rain systems?

p12l3f: I don't agree with that. the method has problems with, heavy rain (cold, tall

clouds), with complex rain systems? and in winter

---

## Referee Comment (RC2) · Anonymous Referee #2 · 30 Mar 2018

The authors proposed a downscaling method based on CDF to obtain hourly 0.05 ° grid precipitation data. This topic is interesting and would be useful for the climate change community. In general, this paper is well-written for most parts. However, a minor revision is needed before it is published in HESS.

Comments:

1. Precipitation is more complex to downscaling than temperature. Until now, hundreds of methods have been developed via statistical and dynamical approaches. However, there is none common method for all regions. The authors did not state clearly about statistical and dynamical downscaling methods in the introduction. The presented study is a statistical downscaling which only used the outputs (CMORPH and GEO-IR) to explore a statistical link. From my side, this approach is similar with Quantile-Mapping (QM), which the authors did not mention. What is the difference between QM and DCDF? The reviewer did not figure out from the Equations (1) to (7).

2. Is it possible that the information is missing in the process of 0.05 degree data aggregated to 0.25degree? Can the built relationship from a coarse resolution represent the similar features for a higher resolution? What method was used in the aggregation, sum or mean? Does it affect the result?

3. The structure may be adjusted. I prefer to introduce the data and study area firstly and then followed by the method. The equations for validate criteria are not necessary since they are common used.

4. The authors claimed that DCDF performs better in the frontal rain systems but worse in mountainous. Is CMORPH the main reason for that? If use the reanalysis (e.g. ERA-Interim) for downscaling, will be better? I suggest more discussions on it.

5. How to define the rain-no-rain threshold?

6. How the DCDF works for each region in each month, rather than seasonal? Figure 8 is for all regions?

7. Table 3, CDF => DCDF

8. P7 L18: It seem => It seems

9. What is the specific means of a, b, c, and d in equations 11 to 13.

10. Figure 7 is hard to follow. Please revise it in a more readable way.

11. Some information is missing or wrong in Fig 9a.

---

## Author Comment (AC1) · 21 Apr 2018

**Response to Reviewer #1 comments**

We thank the reviewers for their time in reviewing the manuscript. Those comments are valuable and helpful to improve the manuscript. We have considered the comments very carefully and made revisions to the manuscript. We hope our revision could satisfy your requirements and meet your approval. Our point-by-point replies to the comments and suggestions are described as below.

**Comment 1**

p1l11: "proposed" better present tense?

**Response:**

Thank you for your useful advice. We have revised it. Please see line 11 in page 1 (red color).

**Comment 2**

p2l2: which challenges and problems?

**Response:**

Thank you for your very valuable and careful advice. These problems mean advantages of gauge data and RADAR. We think the description "These problems can be effectively resolved by using satellite remote sensing techniques" may not make logical sense. Thus, we removed this sentence, which will not result in inconsistency of the context.

The main problem is no dense and high-quality gauge network to evaluate RADAR data. Because RADAR has high resolution, there is always no gauge located within the pixel. It is difficult to get the answer how good are RADAR estimates and its full structure of the error distribution.

**Comment 3**

fig1: c is unclear what can be seen there

**Respons**e:

Thank you for your useful advice. We have revised it. Please see figure 2.

**Comment 4**

p3l19ff: the matching is not decribed, what means matching how is it done? Formula?Is that Quantile mapping?

**Response:**

Thank you for your very valuable advice. It is very significant to improve our paper. CDF matching belongs to quantile mapping. CDF matching relates one variable (Tb in our study) to reference (precipitation in our study) using same cumulative frequency. We used figure and formula to explain the CDF matching.

Specifically, the matching process is shown as figure 2 (Reference represents precipitation rate; variable represents Tb). The matching process is implemented by a one-to-one mapping CDF of variable onto that of the reference (Equation 5). We have added the description of the CDF matching in the manuscript and equation 5, and revised figure 2. Please see line 5~6 in page 5 (red color) and figure 2.

The matching process of Tb and CMORPH is decripted in line 11~15 in page 6 (red color).

Thanks again for your valuable advice.

**Comment 5**

p4l11: ..behind [the] downscaling.

**Response:**

Thank you for your useful advice. It has been revised. Please see line 25 in page 5 (red color).

**Comment 6**

p4l28ff: sentence is unclear, what is a non-raining frequency?

**Response:**

Thank you for your very valuable advice. It is very significant to improve our paper. "a non-raining frequency" is an unclear expression. Here, it means the frequency of critical value of rain rate when rain rate is less than the value, it would not rain. As shown in figure below, the rain–no-rain threshold is set at about $v_i$ where the cumulative frequency equals $C_i$.

Specially, all precipitation rate (Tb) are sorted in ascending (descending) order. Then cumulative probability distributions are both obtained. The cumulative probability is defined as critical probability when precipitation rate equals zero. The rain-no-rain threshold is the Tb with cumulative probability same as the critical probability. As shown in Fig. 2c and 2d (T means precipitation rate; V represents Tb), the rain–no-rain threshold is set at about $v_i$ where the cumulative probability equals $C_i$ (critical probability).Please see line 11~15 in page 6 (red color).

Thanks again for your valuable advice.

[Figure]

[Figure]

**Comment 7**

p5l20 and p6l2: A variogram

**Response:**

Thank you for your useful advice. We have revised them. Please see line 8 and line 9 in page 7 (red color).

**Comment 8**

p5l21: (Matheron,..)

**Response:**

Thank you for your useful advice. We have revised it. Please see line 9 in page 7 (red color).

**Comment 9**

Chapter 2.3 and 2.4 also fig 3, tab.1 are well known scores and techniques maybe you skip them.

**Response:**

Thank you for your useful advice. We have removed chapter 2.3, also fig 3 (Schematic of the variogram curve), tab.1 (Contingency table for the definition of the categorical metrics).

**Comment 10**

Fig4: please exclude the islands with climate situations 8 and 9 from the map. The status of these territories are unclear.

**Response:**

We used the distribution of average annual precipitation during 1960~2010 as base map because it is an most important factor for selecting evaluation regions. Please see figure 1.

**Comment 11**

Fig5: is again a processing scheme, maybe it is better to make one out of fig2 and fig5, if you change the order of chapter 2 and 3 you can combine chapters 3.3 and 2.2

**Response:**

Thank you for your so careful and valuable advice. We agreed with you. New figure 3 was made combing fig2 and fig5. We have changed the order of chapter 2 and 3. We first introduced study areas and datasets (chapter 2), and then the methodology (chapter 3). We have combined chapter 3.3 and 2.2 into 3.2.

**Comment 12**

p7: I found to0 much information in chapter 3.2.2 maybe you short it and only refer to the 2 or 3 most important references.

**Response:**

Thank you for your useful advice. We agreed with you. We have removed some redundant description. We think these description are better in discussions. Please see line 3~9 in page 10 (red color).

**Comment 13**

p8l15: [the] image ....definitely needs language editing

**Response:**

Thank you for your so careful and useful advice. We have revised the description. Please see line 4~6 in page 7 (red color).

**Comment 14**

fig7: inscribe, which picture is cmorph data and which dcdf, and which picture belongs to which region, maybe confusing or unclear for the reader.

**Response:**

Thank you for your useful advice. We have revise figure5. Please see fig.5.

**Comment 15**

p9l7ff: please write sill, range when necessary instead of d and c+c0

**Response:**

Thank you for your useful advice. We have revised them. Please see line 7, 8 an 15 in page 8 (red color).

**Comment 16**

p9-10l32ff: Fig 10, according to the shown events the conclusion is not significant, the better fit of dcdf at gauge scale may be pure luck.

**Response:**

Thank you for your very valuable advice. It is very significant to improve our paper. It is difficult to validate the representativeness of the selected gauge (point) (red dots in figure below) in every region. We just selected these six gauges because their annual precipitation almost equal to average precipitation over area in respective region.

We have compared the DCDF, CMORPH and gauge for all gauges. You are right that not all the fit of DCDF at gauge scale is better than CMORPH. The result showed that the better fit between DCDF and gauge than that between CMORPH and gauge is 10%. The nearly equivalent fit is 69%. The poorer fit is 21%, and mainly happened in region NW, CW and TP.

We have revised the description in our results (Please see line 32 in page 8 and line 1~5 in page 9) (red color), and conclusions (Please see line 23~24 in page 11) (red color).

Thanks again for your valuable advice.

[Figure]

**Comment 17**

p10l11ff: the bad performance of the approach in winter was something i except reading your methods. How are the correlations between tb and rain in teh winter months?

**Response**:

The table below gives $R^2$ in four seasons. The most average of $R^2$ are higher than 0.90 for six regions in four seasons. The maximum CC is higher than 0.98. Most of the minimum $R^2$ is higher than 0.80 in summer and autumn. Minimum $R^2$ ranges from 0.60 to 0.89 in spring, and from 0.51 to 0.71 in winter. It showed that Tb had relatively poor correlation with precipitation rate in winter. This result may inferred that the bad performance of the approach in winter is mainly caused by low accuracy of CMORPH, which may be also applicable for dry regions and mountainous or hilly areas.

| Time | | SE | CE | NE | CW | NW | TP |
|------|------|------|------|------|------|------|------|
| SP | Mean | 0.91 | 0.97 | 0.96 | 0.98 | 0.97 | 0.98 |
| | Max | 0.99 | 0.99 | 0.99 | 0.99 | 0.99 | 0.99 |
| | Min | **0.64** | 0.89 | **0.60** | 0.83 | **0.73** | **0.78** |
| | Std | 0.05 | 0.02 | 0.04 | 0.01 | 0.02 | 0.02 |
| SU | Mean | 0.92 | 0.96 | 0.96 | 0.97 | 0.99 | 0.97 |
| | Max | 0.98 | 0.99 | 0.99 | 0.99 | 0.99 | 0.99 |
| | Min | 0.84 | **0.77** | 0.85 | 0.86 | 0.97 | 0.86 |
| | Std | 0.03 | 0.03 | 0.02 | 0.03 | 0.00 | 0.03 |
| FA | Mean | 0.97 | 0.97 | 0.97 | 0.97 | 0.88 | 0.98 |
| | Max | 0.99 | 0.99 | 0.99 | 0.99 | 0.99 | 0.99 |
| | Min | 0.82 | 0.89 | 0.87 | 0.86 | **0.64** | 0.94 |
| | Std | 0.04 | 0.03 | 0.03 | 0.02 | 0.11 | 0.01 |
| WI | Mean | 0.92 | 0.92 | 0.89 | 0.95 | 0.92 | 0.97 |
| | Max | 0.99 | 0.99 | 0.99 | 0.99 | 0.99 | 0.99 |
| | Min | **0.65** | **0.51** | **0.60** | **0.71** | **0.58** | **0.69** |
| | Std | 0.07 | 0.07 | 0.09 | 0.04 | 0.07 | 0.03 |

**Comment 18**

p10l13: rain-no-rain threshold, where is this threshold defined? how large is threshold?

**Response**:

Thank you for your very valuable advice. I am sorry I didn't explain it clearly. All precipitation rate (Tb) are sorted in ascending (descending) order. Then cumulative probability distributions are both obtained. The cumulative probability is defined as critical probability when precipitation rate equals zero. The rain-no-rain threshold is the Tb with cumulative probability same as the critical probability. As shown in figure below, the rain–no-rain threshold is set at about vi where the cumulative frequency equals Ci. Please see line 11~15 in page 6 (red color).

The threshold generally ranges from 190K to 270K, and most thresholds fall between 200K and 250K. As examples in fig5, the probability of precipitation rate was the largest for a given Tb in region SE, followed by region CE and then region NE. The

rain-no-rain thresholds for regions CW and NW were approximately 230 K, while 254K for region TP. The probability of precipitation rate was the largest for a given Tb in region TP.

Thanks again for your valuable advice.

[Figure]

**Comment 19**

p10l15: what is meant by complex rain systems?

**Response:**

Thank you for your very helpful advice. I am sorry I didn't describe it exactly. It means orographic rain systems over mountainous or hilly areas. We have revised this sentence. Please see line 16 in page 9 (red color).

**Comment 20**

p12l3f: I don't agree with that. the method has problems with, heavy rain (cold, tall clouds), with complex rain systems? and in winter.

**Response:**

Thank you for your very valuable advice. This description is not accurate. We have revised it. The DCDF reflected more detailed moving and changing processes of rainfall under the condition that DCDF perform better than or nearly equivalent to CMORPH. Please see line 17~18 in page 11 (red color).

---

## Author Comment (AC2) · 21 Apr 2018

**Response to Reviewer #2 comments**

We thank the reviewers for their time in reviewing the manuscript. Those comments are valuable and helpful to improve the manuscript. We have considered the comments very carefully and made revisions to the manuscript. We hope our revision could satisfy your requirements and meet your approval. Our point-by-point replies to the comments and suggestions are described as below.

The authors proposed a downscaling method based on CDF to obtain hourly 0.05 ° grid precipitation data. This topic is interesting and would be useful for the climate change community. In general, this paper is well-written for most parts. However, a minor revision is needed before it is published in HESS.

**Comment 1**

Precipitation is more complex to downscaling than temperature. Until now, hundreds of methods have been developed via statistical and dynamical approaches. However, there is none common method for all regions. The authors did not state clearly about statistical and dynamical downscaling methods in the introduction. The presented study is a statistical downscaling which only used the outputs (CMORPH and GEO-IR) to explore a statistical link. From my side, this approach is similar with Quantile-Mapping (QM), which the authors did not mention. What is the difference between QM and DCDF? The reviewer did not figure out from the Equations (1) to (7).

**Response:**

Thank you for your very valuable advice. It is very significant to improve our paper. CDF matching belongs to quantile mapping. CDF matching relates one variable (Tb in our study) to reference (precipitation in our study) using same cumulative frequency. We used figure and formula to explain the CDF matching.

The matching process is implemented by a one-to-one mapping CDF of variable onto that of the reference (Eq. 5). We added equation 5. Please see line 5~6 in page 5 (red color).

For matching between Tb and CMORPH, all precipitation rate (Tb) are sorted in ascending (descending) order. Then cumulative probability distributions are both obtained. The cumulative probability is defined as critical probability when precipitation rate equals zero. The rain-no-rain threshold is the Tb with cumulative probability same as the critical probability. Then the CDF matching was applied. All pixels in the Tb images (Tb0.05) were divided into two categories, raining ones less than the rain-no-rain threshold and non-raining ones larger than the threshold. Tb-R relationships were applied to these raining pixels. Finally, CMORPH data were downscaled to 1-hour, 0.05°×0.05°.

Two different approaches are currently being pursued. Dynamical downscaling uses regional climate models (RCMs) to translate the large-scale weather evolution from a GCM into a physically consistent evolution at higher resolution. Statistical downscaling is based on empirical relationships between the regional climate and carefully selected large-scale predictor variables. Please see line 19~22 in page 2 (red color).

Thanks again for your valuable advice.

**Comment 2**

Is it possible that the information is missing in the process of 0.05 degree data aggregated to 0.25degree? Can the built relationship from a coarse resolution represent the similar features for a higher resolution? What method was used in the aggregation, sum or mean? Does it affect the result?

**Response:**

Thank you for your very valuable comment. $0.05°$ Tb was aggregated to $0.25°$ by arithmetic averaging. Then $0.25°$ Tb (after aggregation) is matched with CMORPH for raining pixels by quantile-mapping the CDF. Because variability of precipitation cloud is small over $0.25°$ region, we think the values (number=25) of $0.05°$ Tb within a raining pixel are almost the same. That is, the information is to some extent missing in the process of $0.05°$ data aggregated to $0.25°$, but which has a little effect on the Tb-Rain relationship and DCDF result. That is, the built relationship from a coarse resolution generally can represent the similar features for a higher resolution.

We think what method was used in the aggregation would affect the result. If we use the bi-cubic convolution method or bi-linear method, $0.25°$ Tb after aggregation will involves the information beyond $0.25°$ pixel. Then it will have effect on the Tb-Rain relationship and DCDF result, and the built relationship from a coarse resolution generally can not represent the similar features for a higher resolution.

In summary, we think arithmetic averaging method is propably best choice.

**Comment 3**

The structure may be adjusted. I prefer to introduce the data and study area firstly and then followed by the method. The equations for validate criteria are not necessary since they are common used.

**Response:**

Thank you for your helpful advice. We have changed the order of chapter 2 and 3. We first introduced study areas and datasets (chapter 2), and then the methodology (chapter 3). We have removed dedcription of well known validation index (correlation coefficient (CC), root mean square error (RMSE) and bias, and tab.1).

**Comment 4**

The authors claimed that DCDF performs better in the frontal rain systems but worse in mountainous. Is CMORPH the main reason for that? If use the reanalysis (e.g. ERA-Interim) for downscaling, will be better? I suggest more discussions on it.

**Response:**

Thank you for your very valuable advice. We agreed with you. It is very significant to improve our paper. The table below gives R2 in four seasons. The most average of R2 are higher than 0.90 for six regions in four seasons. This result may infer that the bad performance of the approach is mainly caused by low accuracy of CMORPH. Thus using reanalysis data for downscaling may be better than satellite products. Additionally, the assumption of DCDF method is also applied to reanalysis data. It is expected that the DCDF method also applied to reanalysis precipitation data (e.g. ERA-Interim, 0.75°/6 hourly).

We have discussed this issue in the paper. Please see line 13~17 in page 10 (red color).

| Time | | SE | CE | NE | CW | NW | TP |
|------|------|------|------|------|------|------|------|
| SP | Mean | 0.91 | 0.97 | 0.96 | 0.98 | 0.97 | 0.98 |
| | Max | 0.99 | 0.99 | 0.99 | 0.99 | 0.99 | 0.99 |
| | Min | **0.64** | 0.89 | **0.60** | 0.83 | **0.73** | **0.78** |
| | Std | 0.05 | 0.02 | 0.04 | 0.01 | 0.02 | 0.02 |
| | | | | | | | |
| SU | Mean | 0.92 | 0.96 | 0.96 | 0.97 | 0.99 | 0.97 |
| | Max | 0.98 | 0.99 | 0.99 | 0.99 | 0.99 | 0.99 |
| | Min | 0.84 | **0.77** | 0.85 | 0.86 | 0.97 | 0.86 |
| | Std | 0.03 | 0.03 | 0.02 | 0.03 | 0.00 | 0.03 |
| | | | | | | | |
| FA | Mean | 0.97 | 0.97 | 0.97 | 0.97 | 0.88 | 0.98 |
| | Max | 0.99 | 0.99 | 0.99 | 0.99 | 0.99 | 0.99 |
| | Min | 0.82 | 0.89 | 0.87 | 0.86 | **0.64** | 0.94 |
| | Std | 0.04 | 0.03 | 0.03 | 0.02 | 0.11 | 0.01 |
| | | | | | | | |
| WI | Mean | 0.92 | 0.92 | 0.89 | 0.95 | 0.92 | 0.97 |
| | Max | 0.99 | 0.99 | 0.99 | 0.99 | 0.99 | 0.99 |
| | Min | **0.65** | **0.51** | **0.60** | **0.71** | **0.58** | **0.69** |
| | Std | 0.07 | 0.07 | 0.09 | 0.04 | 0.07 | 0.03 |

**Comment 5**

How to define the rain-no-rain threshold?

**Response:**

As shown in figure below, the rain–no-rain threshold is set at about $v_i$ (fig. b) where

the cumulative frequency equals $C_i$ (fig. a and b). Specially, all precipitation rate (Tb) are sorted in ascending (descending) order. Then cumulative probability distributions are both obtained. The cumulative probability is defined as critical probability when precipitation rate equals zero.   The rain-no-rain threshold is the Tb with cumulative probability same as the critical probability. As shown in Fig. 2c and 2d (T means precipitation rate; V represents Tb), the rain–no-rain threshold is set at about vi where the cumulative probability equals Ci (critical probability). Please see line 11~15 in page 6 (red color).

Thanks again for your valuable advice.

[Figure]

**Comment** 6

How the DCDF works for each region in each month, rather than seasonal? Figure 8 is for all regions?

**Response:**

It is unavailable for hourly gauge data. A disdrometer installed at Xingzi station (29.45°N, 116.05°E) in the Jiangxi province (Fig. 1) provided hourly data in 2014, except June and July when the instrument was subject to a transmission error. Figure 8 is just for a point using this disdrometer data at the hourly scale.

Table below lists the statistics (We only showed CC and Bias) at the daily scale for each region in each month. The results showed that DCDF generally performed at each month same as at each season.

| Indexes | Month | Type | SE | CE | NE | CW | NW | TP |
|---|---|---|---|---|---|---|---|---|
| B(%) | Jan | CMORPH | -99.27 | -34.28 | -103.31 | 1294 | 205.31 | 1598 |
| | | DCDF | 18.28 | 27.15 | 22.98 | 2159 | 185.94 | 1628 |
| | Feb | CMORPH | -91.83 | -33.94 | -90.64 | 1323 | 67.24 | 1767 |
| | | DCDF | 5.65 | 15.24 | 20.38 | 1642 | 39.29 | 2338 |
| | Mar | CMORPH | -22.44 | -3.92 | -48.56 | 41.85 | 170.55 | 80.27 |
| | | DCDF | -10.61 | 2.72 | 40.90 | 50.72 | 197.69 | 96.51 |
| | Apr | CMORPH | -20.45 | -3.10 | -43.72 | 45.86 | 153.91 | 82.25 |
| | | DCDF | -10.72 | 3.26 | 37.59 | 44.42 | 192.35 | 100.76 |
| | May | CMORPH | -15.93 | -3.10 | -43.98 | 47.37 | 157.57 | 86.20 |
| | | DCDF | -6.61 | 2.95 | 30.26 | 55.19 | 187.38 | 104.40 |

|     |      |        |        |        |        |        |        |        |
| --- | ---- | ------ | ------ | ------ | ------ | ------ | ------ | ------ |
|     | Jun  | CMORPH | -16.83 | 2.29   | 5.61   | -37.22 | 162.62 | -20.68 |
|     |      | DCDF   | -8.92  | 2.07   | 5.68   | 20.91  | 222.37 | 19.53  |
|     | Jul  | CMORPH | -24.42 | 3.27   | 5.54   | -46.77 | 153.39 | -11.25 |
|     |      | DCDF   | -15.57 | 2.98   | 5.80   | 25.62  | 218.35 | 16.17  |
|     | Aug  | CMORPH | -25.25 | 3.00   | 7.02   | -40.43 | 117.33 | 5.17   |
|     |      | DCDF   | -14.90 | 2.63   | 7.16   | 29.85  | 197.94 | 10.36  |
|     | Sept | CMORPH | -68.59 | -33.92 | -17.58 | 19.90  | 120.25 | 41.33  |
|     |      | DCDF   | 9.39   | 30.28  | 16.68  | 15.47  | 118.37 | 54.45  |
|     | Oct  | CMORPH | -69.99 | -34.74 | -19.21 | 9.18   | 118.69 | 45.95  |
|     |      | DCDF   | 11.57  | 36.56  | 12.70  | 17.37  | 130.66 | 58.72  |
|     | Nov  | CMORPH | -50.36 | -27.92 | -12.15 | 11.29  | 112.30 | 48.99  |
|     |      | DCDF   | 0.21   | 39.88  | 20.33  | 9.86   | 116.71 | 62.28  |
|     | Dec  | CMORPH | -95.41 | -30.92 | -91.95 | 921    | 61.25  | 1770   |
|     |      | DCDF   | 7.73   | 22.2   | 23.51  | 1623   | 50.22  | 1862   |
| CC  | Jan  | CMORPH | 0.30   | 0.01   | 0.00   | 0.06   | 0.03   | 0.05   |
|     |      | DCDF   | 0.45   | 0.16   | 0.04   | 0.01   | 0.02   | 0.11   |
|     | Feb  | CMORPH | 0.38   | 0.05   | 0.00   | 0.06   | 0.04   | 0.10   |
|     |      | DCDF   | 0.60   | 0.18   | 0.06   | 0.01   | 0.04   | 0.19   |
|     | Mar  | CMORPH | 0.57   | 0.31   | 0.36   | 0.20   | 0.07   | 0.00   |
|     |      | DCDF   | 0.63   | 0.39   | 0.35   | 0.16   | 0.06   | 0.01   |
|     | Apr  | CMORPH | 0.61   | 0.42   | 0.38   | 0.17   | 0.07   | 0.06   |
|     |      | DCDF   | 0.64   | 0.41   | 0.38   | 0.19   | 0.05   | 0.05   |
|     | May  | CMORPH | 0.67   | 0.34   | 0.36   | 0.17   | 0.07   | 0.09   |
|     |      | DCDF   | 0.71   | 0.45   | 0.38   | 0.17   | 0.05   | 0.08   |
|     | Jun  | CMORPH | 0.38   | 0.26   | 0.30   | 0.17   | 0.41   | 0.30   |
|     |      | DCDF   | 0.49   | 0.28   | 0.51   | 0.46   | 0.44   | 0.39   |
|     | Jul  | CMORPH | 0.36   | 0.17   | 0.24   | 0.17   | 0.40   | 0.22   |
|     |      | DCDF   | 0.47   | 0.27   | 0.44   | 0.44   | 0.45   | 0.35   |
|     | Aug  | CMORPH | 0.35   | 0.18   | 0.24   | 0.17   | 0.40   | 0.22   |
|     |      | DCDF   | 0.47   | 0.25   | 0.43   | 0.44   | 0.44   | 0.36   |
|     | Sept | CMORPH | 0.39   | 0.48   | 0.36   | 0.07   | 0.31   | 0.19   |
|     |      | DCDF   | 0.54   | 0.50   | 0.44   | 0.09   | 0.20   | 0.08   |
|     | Oct  | CMORPH | 0.41   | 0.48   | 0.36   | 0.07   | 0.32   | 0.10   |
|     |      | DCDF   | 0.51   | 0.52   | 0.50   | 0.09   | 0.20   | 0.08   |
|     | Nov  | CMORPH | 0.42   | 0.54   | 0.36   | 0.07   | 0.35   | 0.06   |
|     |      | DCDF   | 0.52   | 0.54   | 0.51   | 0.14   | 0.26   | 0.08   |
|     | Dec  | CMORPH | 0.33   | 0.02   | 0.00   | 0.05   | 0.02   | 0.06   |
|     |      | DCDF   | 0.47   | 0.16   | 0.06   | 0.00   | 0.02   | 0.17   |

**Comment 7**

Table 3, CDF => DCDF

**Response:**

Thank you for your useful advice. We have revised them. Please see table 2.

**Comment 8**

P7 L18: It seem => It seems

**Response**:

Thank you for your useful advice. We have revised them. Please see line 3 in page 10 (red color).

**Comment 9**

What is the specific means of a, b, c, and d in equations 11 to 13.

**Response**:

Thank you for your useful advice. These evaluation metrics in chapter 2.3 and 2.4 are well known, thus they have been deleted. We have removed chapter 2.3 and 2.4 also fig 3 (Schematic of the variogram curve), tab.1 (Contingency table for the definition of the categorical metrics).

**Comment 10**

Figure 7 is hard to follow. Please revise it in a more readable way.

**Response**:

Thank you for your useful advice. We have revised it. Please see fig. 5.

**Comment 11**

Some information is missing or wrong in Fig 9a.

**Response**:

Thank you for your useful advice. We have revised them. Please see line Fig 7a.

---

## Author Comment (AC3) · 21 Apr 2018

**Precipitation downscaling using a probability-matching approach and geostationary infrared data: An evaluation over six climate regions**

Ruifang Guo1,2, Yuanbo Liu1, Han Zhou1,2, Yaqiao Zhu3

1Key Laboratory of Watershed Geographic Sciences, Nanjing Institute of Geography and Limnology, Chinese Academy of Sciences, No. 73 East Beijing Road, Nanjing 210008, China
 2University of Chinese Academy of Sciences, No. 19 Yuquan Road, Beijing 100049, China
 3College of Urban and Environmental Sciences, Hubei Normal University, No.11 Cihu road, Huangshi 435002, China

\*Correspondence to: Yuanbo Liu (ybliu@niglas.ac.cn)

- 10 Abstract. Precipitation is one of the most important components of the global water cycle. Precipitation data at high spatial and temporal resolutions are crucial for basin-scale hydrological and meteorological studies. In this study, we propose a cumulative distribution of frequency (CDF)-based downscaling method (DCDF) to obtain hourly 0.05°×0.05° precipitation data. The main hypothesis is that a variable with the same resolution of target data should produce a CDF that is similar to the reference data. The method was demonstrated using the 3 hourly 0.25°×0.25° Climate Prediction Center Morphing
- 15 method (CMORPH) dataset and the hourly 0.05°×0.05° FY2-E Geostationary (GEO) Infrared (IR) temperature brightness (Tb) data. Initially, power function relationships were established between precipitation rate and Tb for each 1°×1° region. Then the CMORPH data were downscaled to 0.05°×0.05°. The downscaled results were validated over diverse rainfall regimes in China. Within each rainfall regime, the fitting functions coefficients were able to implicitly reflect the characteristics of precipitation. Quantitatively, the downscaled estimates not only improved spatio-temporal resolutions, but
- 20 also performed better (Bias: -7.35%~10.35%; correlation coefficient (CC): 0.48~0.60) than the CMORPH product (Bias: 20.82%~94.19%; CC: 0.31~0.59) over convective precipitating regions. The downscaled results performed as well as the CMORPH product over regions dominated with frontal rain systems and performed relatively poorly over mountainous or hilly areas where orographic rain systems dominate. Qualitatively, at the daily scale, DCDF and CMORPH had nearly equivalent performances at the regional scale, and 79% DCDF may perform better than or nearly equivalent to CMORPH at
- 25

30

under the condition that DCDF perform better than or nearly equivalent to CMORPH.

**1** Introduction**

Precipitation is a critical component in the global water cycle (Barrett and Martin, 1981; Smith et al., 1998; Tobler, 2004). Precipitation data at spatio-temporal resolutions are favoured mainly for two reasons. First, the poor representativeness and unevenly distribution of gauge stations make it incapable of reflecting spatially the precipitation variations (Hughes, 2006,

the point (rain gauge) scale. The downscaled estimates were able to capture more details about rainfall motions and changes

Collischonn et al., 2008; Javanmard et al., 2010). Second, ground radar systems can provide full coverage spatial data for most regions, but RADAR is very week in view of the precipitation intensity and subject to short time series. Moreover, the validation poses a big challenge for hydrological applications (Krajewski and Smith, 2002).

5

10

A number of techniques have been developed to estimate or retrieve precipitation (Kidd and Levizzani, 2011). Based on these technologies, precipitation datasets have been produced at various resolutions, including the Global Precipitation Climatology Project (GPCP) (Huffman et al., 1997, 2001, 2009), the Tropical Rainfall Measuring Mission (TRMM) Multi-Satellite Precipitation (TMPA) (Huffman et al., 2007), the Climate Prediction Center Morphing method (CMORPH) (Joyce et al., 2004) and the Global Satellite Mapping of Precipitation (GSMaP) (Ushio et al., 2009), especially over the last 20 years. The typical spatial resolution of these products is  $0.25^{\circ} \times 0.25^{\circ}$  (Dinku et al., 2007; Ebert et al., 2007; Hirpa et al., 2010; Sohn et al., 2010; Bitew and Gebremichael, 2011; Romilly and Gebremichael, 2011; Thiemig et al., 2012; Hu et al., 2014). This coarse resolution generally impedes the applications of the data for basin-scale hydrological and meteorological studies (Mekonnen et al., 2008). A downscaling procedure would therefore be highly necessary to meet the requirements of smallscale (

$$Tb_{l} = \frac{1}{n} \sum_{i=0}^{n} Tb_{h}(i)$$
(6)

$$Tb_l = m \times R_l^{\ e} \tag{7}$$

$$R_h = \left(\frac{Tb_h}{m}\right)^{\frac{1}{e}} \tag{8}$$

where Tbh denotes high-resolution GEO-IR Tb data. Tb1 denotes upscaled Tb data. R1 denotes the low-resolution
precipitation product. Rh denotes the derived high-resolution estimates. m and e are coefficients of the Tb-R relationship, and n is the number of high-resolution pixels within a low-resolution pixel.

Under the assumption that colder clouds are linked to higher rainfall than warmer clouds, the downscaling method assumes a monotonically increasing precipitation rate with decreasing Tb. Therefore, cumulative histograms of precipitation rate and Tb are matched, so that the occurrence of the heaviest precipitation is associated with the Tb values linked to the heaviest rainfall. Decreasing Tb values are assigned to increasing precipitation rates so that the final distribution of Tb assigned to the precipitation rates is the same as that determined using precipitation rate data. Specially, all precipitation rate (Tb) are sorted in ascending (descending) order. Then cumulative probability distributions are both obtained. The cumulative probability is defined as critical probability when precipitation rate equals zero. The rain-no-rain threshold is the Tb with cumulative probability same as the critical probability. As shown in Fig. 2c and 2d (T means precipitation rate; V represents

15 Tb), the rain–no-rain threshold is set at about  $v_i$  where the cumulative probability equals  $C_i$  (critical probability).

The specific steps used for downscaling with CMORPH and FY2-E IR data are described as follows:

a. Aggregate IR-Tb data (Tb0.05) from 0.05° to 0.25° by pixel averaging (Tb0.25).

IR-Tb data (Tb0.05) were aggregated to a  $0.25^{\circ}$  grid (Tb0.25) for each 3-hour period (0000-0300, 0300-0600, . . ., 2100-2400 UTC), in order to match the spatial and temporal resolutions as CMORPH.

b. Generate the histogram database for CDF matching.

IR-Tb (Tb0.25) and CMORPH precipitation rate ( $R_{0.25}$ ) were recorded in a database. The sample area for CDF matching was determined as follows. The horizontal and temporal scales of stratiform precipitation range from 101 to 103 kilometers and from hours to days (Orlanski, 1975; Trapp, 2013), while those of cumuliform precipitation range from a few km to tens of kilometers and from minutes to hours (Orlanski, 1975; Rickenbach, 2008). In combination with previous studies (Kidd et

25 al., 2003; Huffman et al. 2007), the downscaling procedure was conducted at 1°×1° grids over a 10-days period. To reduce the heterogeneity among grids, a 3×3 window was used for smoothing purpose.

c. Build relationships between precipitation rate and Tb

The histograms of Tb and precipitation rate were generated and converted to cumulative histograms, and then matched using the CDF matching (As shown in fig. 1. precipitation rate means T; Tb represents V;  $v_i$  is the rain-no-rain threshold). Power function relationship between precipitation rate ( $R_{0.25}$ ) and Tb ( $Tb_{0.25}$ ) was established for each 1°×1° area over a 10-

30

10

20

days period. Meanwhile, various parameters, including coefficients of the Tb-R relationship, rain-no-rain threshold and R2, were obtained.

d. Estimate precipitation rate pixel by pixel at 1-hour, 0.05°

All pixels in the Tb images (Tb0.05) were divided into two categories, raining ones less than the rain-no-rain threshold and non-raining ones larger than the threshold. Tb-R relationships were applied to these raining pixels. Finally, CMORPH data were downscaled to 1-hour,  $0.05^{\circ} \times 0.05^{\circ}$ .

**3.3 Variogram**

5

10

A variogram describes how data correlates with distance. The variogram function  $\gamma(h)$  is defined as half of the mean value of the square of the difference between points separated by a distance h (Matheron, 1963). A variogram is generally an increasing function of distance h. The relationship between  $\gamma(h)$  and h is commonly described using the nugget effect (C0), sill (C0+C) and range (D). C0 denotes micro-scale variations, equated to of  $\gamma(0)$ . C0+C denotes limit of the variogram  $\gamma$  (+ $\infty$ ). D denotes the distance at which the difference of the variogram from the sill becomes negligible. Variogram is used here to describe the spatial structure of satellite precipitation data.

**4** Results**

**15 4.1 Tb-precipitation rate relationship**

precipitation characteristics well.

[revised manuscript text omitted]

---

## Author Comment (AC4) · 21 Apr 2018

The comment was uploaded in the form of a supplement:
https://www.hydrol-earth-syst-sci-discuss.net/hess-2017-592/hess-2017-592-AC4-supplement.pdf